# Health Issues Due to the Global Prevalence of Sedentariness and Recommendations towards Achieving a Healthier Behaviour

**DOI:** 10.3390/healthcare9080995

**Published:** 2021-08-04

**Authors:** Cédrick T. Bonnet, Jose A. Barela

**Affiliations:** 1Univ. Lille, CNRS, UMR 9193–SCALab–Sciences Cognitives et Sciences Affectives, F-59000 Lille, France; 2Institute of Biosciences, São Paulo State University, Rio Claro 13506-900, Brazil; jose.barela@unesp.br

**Keywords:** prolonged sedentariness, non-communicable disease, recommendations, frequent alternance sitting/standing, benefits of the standing position

## Abstract

Sedentariness has progressed in recent years. Here, we summarize the high prevalence of objectively measured sedentariness and the list of health problems associated with sedentariness. According to the literature, a minimum sedentary time of 8 h/d may avoid the harmful effects of sedentariness. Our review of the literature shows that many countries worldwide exceed this threshold. The coronavirus disease 2019 pandemic has increased the proportion of time spent seated in chairs and/or other types of furniture. Furthermore, prolonged sedentariness will continue to increase because it is assumed that people, at least those in desk jobs, perform their work better when sitting than when standing. Many practical solutions should be implemented to help people reduce their sedentary time. People need to be aware that prolonged sedentariness causes health problems. They need to measure the amount of time spent being sedentary to self-guide their behaviour. They should adopt a new lifestyle to avoid prolonged sedentariness and prolonged standing. In addition, we point out that they should frequently change their posture to avoid fatigue and health issues. For global public health, there is an urgent need to adopt an intermediate healthy/healthier behaviour between too much time spent in the sitting and standing positions.

## 1. Introduction

Non-communicable diseases associated with lack of physical activity have been thoroughly documented over the last few decades [1,2,3]. More light should be shed on the effects of sedentariness because non-communicable diseases associated with sedentariness are more recently documented [4,5,6]. Here, sedentariness is defined as any waking behavior characterized by an energy expenditure ≤1.5 metabolic equivalents, while in a sitting, reclining or lying posture [7]. In fact, the level of sedentariness is increasing worldwide at least since the middle of the 20th century [8]. Furthermore, the coronavirus disease 2019 (COVID-19) pandemic and the resulting structural changes in society (i.e., more teleworking, videoconferencing, and online shopping) [9] have prompted people to become more sedentary and will continue to do so. 

In the first section of this article, which is related to the healthcare policy, we highlight that since the end of World War II, the prevalence of sedentariness has increased [8]. We obtained evidence that people in high-income countries, and especially those in desk jobs, spend long hours sitting, which is known as prolonged sedentariness. We review the list of non-communicable diseases associated with prolonged sedentariness. The latest epidemiological data highlight the need for clear proposals on how to reverse sedentariness [4,5]. In the second section of this article, we highlight that spending more time in the standing position (simply standing, walking, performing any kind of other motor action) or doing any physical activity even when sitting (cycling, working with a pedal workstation) during the day are good solutions to spending less time in sedentariness. We also explain that the recommendation to spend more time in the standing position should be observed with caution to avoid prolonged standing, which is also a well-documented dangerous behaviour [10,11]. Therefore, the time spent in a standing position should not be abruptly increased, but only gradually increased over time. To increase the time spent in the standing position and avoid physical injuries caused by too much time spent standing, it is recommended to frequently alternate standing and sitting positions during the day [12]. We also recommend using a device to objectively measure, monitor, and report the time spent in sedentariness. The additional implementation of these recommendations should promote healthier behaviours. We need to be clear that our message is general as we only discuss the total amount of time spent seated per day without differentiating the context in which people sit (e.g., at work, at home in front of the TV). Moreover, the recommendations are general as they are not specific to any group of individuals (children, adults, older adults, individuals with impairment).

## 2. Sedentariness as a Subconscious, Dangerous, Non-Communicable Disease

Research on modern-day hunter-gatherers suggests that our ancestors spend as much time in inactivity as we do today [13]. However, these hunter-gatherers spend more time in dynamic nonambulatory behaviours, such as squatting and kneeling [13]. By contrast, many humans living in modern societies spend much time in passive nonambulatory behaviour, such as sitting on chairs and/or other human-made furniture [14,15]. Since World War II, technological development and innovation have quickly accentuated sedentariness [8]. Since the end of the 20th century, people sit when using computers, the Internet, and entertainment technologies, and when engaged with display screens of all types [8,16,17]. These technological developments and innovations explain why sitting became the dominant position at work and at home. In fact, many objective measures of sedentariness (using accelerometers and/or inclinometers) performed in many high-income countries showed that healthy adults (aged 18–65 years) spend between 50% and 75% of their waking time in sedentariness [14,18,19,20,21,22,23,24]. Overall, the average proportions of time spent seated in 8 studies are greater than 8 h/d (Table 1, Figure 1) [14,18,19,20,21,22,23,24]. Furthermore, these previous average values are underestimated because the participants did not wear the measurement device during all waking hours. The COVID-19 pandemic has also substantially increased the proportion of time spent seated [9,25]. Moreover, people working in desk jobs spend even more time in sedentariness (from 65% to 82%, depending on the study) [15,26,27]. Sedentariness is unlikely to decrease because of the current COVID-19 pandemic [9].

Schematic representation of the general message that humans should exhibit intermediate behaviors to avoid non-communicable disease caused by too much time spent in sedentariness and physical problems caused by too much time spent in the standing position. The recommended threshold (8 h/d) is very general and conservative, and it should be adjusted according to the type of people (children, healthy adults, older adults, persons with impairments) and to the type of behavior (sitting at work, sitting in front of a TV, performing physical activity during the day). The black line shows that the time spent by people in so many countries is at least greater than 8 h/d (cf our review). Below this black line in Figure 1, a threshold for too much time spent in the standing position is also drawn. This threshold also really depends on the type of people and activity performed during the day. The present manuscript does not discuss this threshold but suggests that this maximum time spent in the standing position should also exist. We used the approximation of 16 h/d for the waking day.

Certainly sedentariness remains a critical problem. Indeed, people can perform a variety of tasks while sitting. They can work, organize their daily schedules, and get anything they wish without moving. They are, or can be, seated at work (workplaces, meeting rooms, conference rooms, and classrooms), when attending various entertainment activities (concerts, cinemas, music lessons, TV, and video games), when commuting (bus, metro, train, and car), and during waiting times (waiting rooms). Moreover, it is typically assumed that people perform better and are more productive when seated than when standing [12]. Many workers worldwide are so inclined to perform their work well that solutions focused solely on health are only considered to be secondary. Furthermore, teleworking provides many advantages such as increase in work productivity, increase in the safety of employees, and reduction of pollution [9]. Ng and Popkin [8] showed the potential increase in the time spent in sedentariness in the next 9 years, at least until 2030. Therefore, effective solutions should be pursued to counter this worldwide unhealthy general trend.

As a primary problem, prolonged sedentariness is considered to be a non-communicable disease that is responsible for at least 35 disorders and diseases [6,22]. Indeed, prolonged sedentariness is proven to cause premature mortality [16,28], type 2 diabetes [17,29], cancer [16,29], vascular and cardiovascular disease [16,28,29,30], stroke [31], chronic inflammatory diseases [32], musculoskeletal disorders [29], poor muscle endurance and loss of force [32], overweight and obesity [16,17], sleep disorders [17], and immune and endocrine impairments [32]. Prolonged sedentariness also impairs an individual’s executive function, cognitive performance, decision-making ability [32], and self-control [32]. Prolonged sedentariness is associated with anxiety [17,31], demotivation, discomfort [33], and increased depression [17]. It also promotes orthostatic intolerance, which is defined as a sustained reduction in arterial blood pressure [32]. Prolonged sedentariness leads to the impairment of various cells, tissues, skeletal muscles, and metabolic flexibility [32], also of lipolysis and blood glucose regulation [6,34]. These problems interact with one another. For example, people who spend more time being sedentary have lower levels of social interactions [31]. Less social interactions can increase the levels of anxiety, which in turn can lead to the development of psychological and cognitive disorders [17,31]. Sitting behaviour is addictive [35]; that is, sedentariness decreases the desire to stand [32] and therefore creates a vicious circle. Hence, prolonged sedentariness is assumed to be a central disease [32], which originated from other problems and disorders [6]. Therefore, measures to prevent the negative health consequences of prolonged sedentariness are urgently needed, especially during this pandemic period. We need to emphasize that modern-day hunter-gatherers are almost not affected by any of these non-communicable diseases [13]. As mentioned above, modern-day hunter-gatherers spend as much time in nonambulatory behaviour [13]. However, they spend more time in dynamic nonambulatory behaviours and therefore less time in passive nonambulatory behaviours, i.e., sitting, than modern humans. This change in dynamic vs. static nonambulatory behaviours may be a key issue to understanding the existence of non-communicable diseases [13]. The physiology of human body is not adapted to spend too many hours in sedentariness [13]. If too many hours are spent in sedentariness every day, the body physiology may be disrupted over time. 

Sedentariness differs from physical inactivity. Indeed, people need to move to interrupt their physical inactivity while simply standing can already interrupt sedentariness. Sedentariness and physical inactivity interact and contribute to the development of non-communicable diseases. In fact, there is a negative association between physical activity and prolonged sedentariness as non-communicable diseases occur when people are both inactive and spend too much time in sedentariness [16,28]. However, there is no positive association between physical activity and less time spent in sedentariness as people can exercise a lot but still spend much time in sedentariness [36,37]. Moreover, people develop the same non-communicable diseases associated with prolonged sedentariness [17,31] even if they are physically active. The only exception is that physical activity can reduce non-communicable diseases due to prolonged sedentariness when the level of physical activity is extremely high (60–75 min/d of moderate-intensity physical activity) [16,28]. Therefore, non-communicable diseases due to prolonged sedentariness cannot be avoided even if light or moderate physical activity is performed, as a higher level of exercise is needed. Instead, an efficient way to avoid non-communicable diseases due to prolonged sedentariness is to interrupt and avoid prolonged sedentariness. 

## 3. Recommendations for Public Health Implications

The WHO [38] already synthetizes that worldwide time spent in sedentariness is too high and urged individuals to reduce this level of sedentariness, giving specific recommendations for various groups of person (e.g., children, adults, older adults, persons with impairments). At the scientific level, our review and analysis indeed confirmed that people worldwide objectively spend too much time in prolonged sedentariness, potentially causing harmful effects on their health. Many studies have used a threshold of 6 to 8 h/d to examine the association between prolonged sedentariness and non-communicable diseases in healthy adults [19,28,39,40,41,42]. These studies found significant associations between sedentariness and non-communicable diseases; that is, prolonged sedentariness could cause non-communicable diseases in these originally healthy adults. Hence, according to this literature, [19,28,39,40,41,42] a general conservative minimum threshold of sedentariness to avoid its harmful effects could be 8 h/d (Figure 1). Here, we only suggest that spending more than 8 h/d in sedentariness may be dangerous if this behaviour is a long-lasting lifestyle habit. However, we do not suggest that spending less than 8 h/d in sedentariness is not harmful. Even spending 7 h/d or 6 h/d in sedentariness could be harmful, as suggested by many studies discussing non-communicable diseases caused by sedentary behaviour such as watching TV [28,43].

As part of the efforts to promote public health, people should be informed about this general and conservative threshold (8 h/d), to avoid the harmful effects of sedentariness [4]. Indeed, if people are not informed of the negative health consequences of sedentariness for more than 8 h/d, they most likely will not be motivated to change their behaviour. If they do not know that sitting is considered a disease [6,32], they most likely will not be encouraged to change their behaviour. In practical terms, assuming that the waking day is 16 h, people should stand and/or spend time in dynamic nonambulatory position (e.g., squatting and kneeling) and/or dynamic sitting (e.g., on a big ball or on a stool) for at least 8 h/d. Otherwise, they would spend more than 8 h/day in sedentariness. The conservative threshold of 8 h/d is easy to remember as it is half of a conventional waking day of 16 h/d (Figure 1), it is one third of a full day. This threshold is merely a general minimum threshold of sedentariness to be re-adjusted in future studies, in the same way as the threshold for physical activity was suggested and then re-evaluated [38]. 

We also encourage people to objectively monitor their time spent in sedentariness. Indeed, people do not accurately report the time they spend in sedentary behaviour [5,22,44]. Self-reports are not reliable in terms of accuracy of the measure and variability of the errors, and these values do not correlate with the objective values [44]. People clearly underestimate the time they spend in sedentariness as showed by many studies [5,22,44]. It important to mention that we are not aware of the existence of any commercially, user-friendly, highly accurate device for objectively measuring sedentariness in the population. To our knowledge, currently available devices (such as accelerometers and inclinometers) [45] are expensive, difficult to use, and are mainly applicable in research studies. This shortcoming is unexpected and surprising, especially when people are being broadly encouraged to use information technology and electronic devices to self-monitor other aspects of their health [45]. Today, we are aware of smart watches and mobile applications to signal if individuals spend too much time on sedentariness [46,47]. This is a good start to break prolonged sedentariness, to suggest standing and moving around, or to begin a short period of time working or performing other tasks while in the standing position. However, both smart watches and mobile phones are limited as they do not measure and report the total time spent in sedentariness during the day. Novel solutions are appearing in the literature but are still in development [48]. These devices do not measure sedentariness accurately because people do not carry their cell phone at each moment all day long. Moreover these devices are not perfectly accurate in their measure of the time spent in sedentariness. One solution was proposed by Gill et al. [49], but this smart watch is still not commercially available today.

Given that many populations worldwide are spending more than 8 h/d in sedentariness (Figure 1) [14,18,20,21,22,23,24], it is clear that spending more time in the standing position could be beneficial. The question then holds as to whether people could be as efficient and productive at work when standing than when sitting. If people were more efficient and productive in the sitting position, it would be difficult to accept the solution to work more often in the standing position. In fact, many recent studies have shown and explained that spending more time in the standing position does not negatively impact performance of tasks and productivity of individuals working in desk-based jobs [33,50]. According to this information, people do not have to sit to be more effective and productive at work. The perspective of performing work satisfactorily while in the standing position is not intuitive but reflects new discoveries [33,50,51,52]. Hence, we encourage future studies to continue to show that more time spent in the standing position at work could facilitate good performance and work productivity. 

In general, people do not spend enough time in the standing position either in nonambulatory or ambulatory behaviours (Figure 1). Therefore, our recommendation is to find solutions to spend more time in the standing position during the day. This recommendation should be considered carefully and requires clarification and explanation. First, it is not applicable to everyone. It should be adapted for people with disabilities and/or difficulties to stand [53]. For healthy adults, considering a waking day of 16 h/d, any one spending less than 6 h/d in sedentariness may not need so much to increase the time spent in the standing position (Figure 1). Behavioural habits should primarily change in people who spend less than 8 h/day in the standing position and/or in dynamic nonambulatory behaviours. Second, sedentary behaviours should be changed gradually instead of abruptly, in the same way as physical activity should be changed gradually for those who do not meet WHO recommendations [38]. For example, a person who spend more than 12 h/d in sedentariness would clearly need to increase the time spent in the standing position and/or dynamic nonambulatory behaviours. Otherwise, this person is exposed to non-communicable diseases (Figure 1), as already mentioned earlier [6,16,17,28,32]. For this person or any other person, the recommendation would not be to abruptly change from prolonged sedentariness to prolonged standing. Indeed, prolonged standing is clearly acknowledged to be harmful for global health and work productivity [10]. In fact, Baker et al. [11] showed that one additional hour standing per day might already cause health problem. For this person, the recommendation would be to slowly change from prolonged sedentariness to a more intermediate behaviour with more time spent in the standing position either in nonambulatory or ambulatory behaviours. 

People who have intermediate behaviour could be referred to as people who are not engaged in prolonged sedentariness and/or prolonged standing but who still spend enough time in both behaviours during the waking day (Figure 1). To the best of our knowledge, this notion of “intermediate behaviour” is new. In its absolute sense, the term refers to sitting less than 8 h/d (Figure 1). At the individual level, a strategy to achieve this recommended threshold needs to be adapted. For example, in order for person A to avoid spending 10 h/d in sedentariness, the strategy to adopt the intermediate behaviour could be to sit less from one period to another, with the change implemented gradually [12]. In order for person A to achieve this intermediate behaviour, an appropriate strategy would be to increase the time spent in the standing position by 30–40 min during a period and then increase again by 30–40 min as discussed in the previous paragraph. This person A should do so until sitting at least less than 8 h/d. Hence, it would require many months and even years of practice for this person to develop and adopt the intermediate behaviour. This recommendation to spend 30 to 40 additional min/d can be beneficial if office workers stand additional 5 min per hour during their day at work [11]. Meanwhile, in order for person B used to stand 7 h/d every day, no change in lifestyle habits would be required, as this individual would already use the intermediate behavior. However, this person B could still reduce his/her time spent in sedentariness in sitting less than 7 h/d, e.g., 6 h/d, to move further away from the threshold of 8 h/d. For this second person B, the objective to stand less than 6 h/d would only require a few months of practice.

We further recommend to frequently alternate between standing and sitting, that is standing up and standing very often in short periods of time instead of once in a long period of time. People must change their behaviour not only for some weeks but as a new lifestyle habit. They should gradually shift towards spending more time in the standing position and/or dynamic nonambulatory behaviours. In alternating between standing and sitting, people could reduce the risk of developing non-communicable diseases caused by prolonged sedentariness [30] and also physical problems caused by prolonged standing. As suggested in the previous paragraph, standing up for 5 additional min every hour at work has been recommended [12]. The ultimate goal of this strategy is to adopt a healthier, intermediate behaviour to reach an intermediate behaviour (Figure 1). To reach this intermediate behaviour, we also recommend standing when calling someone or to hold stand-up meetings over standing only once for a longer period (e.g., 1 additional h/d). In fact, any behaviour (standing or sitting) used in excess does not promote global health; it is better to alternate both behaviours throughout the day. Many other solutions exist to increase the time spent in the standing position [54]. Some examples of intervention techniques are “prompting”, “social influence”, “feedback”, and “anchoring” [54,55]. Prompting refers to messages (posters, mobile phone messages, alert on a phone) engaging people to use the stairs instead of the elevators, to actively move to colleagues’ desk instead of emailing or calling them. Social influence refers to messages (oral, written) that specify what others have done (e.g., number of time other colleagues have taken the stairs during the day) to induce the same behaviors in all workers. Feedback refers to information of any behaviour to find out if one respects expected results, either in front of one’s personal objectives or related to another’s. Anchoring refers to challenges suggested to workers, e.g., to assign a 5000 steps objective during the day or to stand more than 50 times during the day. Venema et al. [56] discussed that most of the office workers were favorable about any intervention, even large ones, if these changes could improve their health. Only 11% disapproved the nudge intervention in their study [56]. In trains, buses, metros or when waiting at some places (for example in town for an appointment), people could stay in the standing position instead of sitting. In classrooms and/or universities, students could listen to professors sometimes upright, sometimes seated [57]. All these small changes would be appropriate to increase the time spent in the standing position to come back to a healthy/healthier intermediate behavior (Figure 1). 

## 4. Conclusions

With regard to sedentariness, the world is not moving toward a desirable and healthier goal. Indeed, more than half of the population worldwide is already engaged in too much time spent in sedentary unhealthy behaviour (Figure 1). Again, human body and physiology is not adapted to such excessive sedentariness. It means that individuals should stand or perform dynamic nonambulatory activities at least half of their waking day (Figure 1). We claim that humans need to adopt new lifestyles and find solutions to reduce prolonged sedentariness to come back to a healthy intermediate behaviour. Important is to acknowledge that non-communicable diseases were the leading cause of death worldwide already in 2010 and that sedentary behaviour is one of the major causes of non-communicable diseases [58]. One solution to reduce the time spent in sedentariness is to spend more time in dynamic nonambulatory (e.g., standing, sitting on unstable surface such as big bowls, squatting, kneeling) behaviours during the waking day. Another solution to spend more time in the standing position during the day is to spend more time in dynamic ambulatory behaviour (e.g., walking, going up and down the stairs, running, performing any kind of physical activity). As we pointed out and need to recall here, the increase time spent in the standing position per day should not be abruptly increased, but only gradually increased over time. In the present manuscript, we explained, the best lifestyle recommendation to frequently alternate sitting and standing positions in order to balance time spent seated and standing, in order to reach a healthier intermediate behaviour (Figure 1). This new lifestyle is possible to achieve, but will require more use of sit-stand desks and devices to objectively measure and determine the proportion of time spent seated. Indeed, people spend much time at work during the week and our societies are built based on the idea of performing most of the activities in a sitting position. Also at home, people do not have always something to do and can come to watch TV or their computer for long hours. They thus need to verify their total time spent seated during the day to regulate their behaviour.

In our manuscript, we are aware that we did not provide any specific minimum threshold for each type of people (infants, healthy adults, older adults, persons with disabilities) and/or for any individual in particular. Future studies should carefully analyse, display and then adjust these respective minimum thresholds. Furthermore, the population needs to be informed and be aware of these thresholds and, more importantly, be motivated to become active in preventing, at least limiting, non-communicable diseases related to prolonged sedentariness. Our main message is that everybody should be careful about adopting an intermediate healthy—at least healthier—behaviour (Figure 1).

## Figures and Tables

**Figure 1 healthcare-09-00995-f001:**
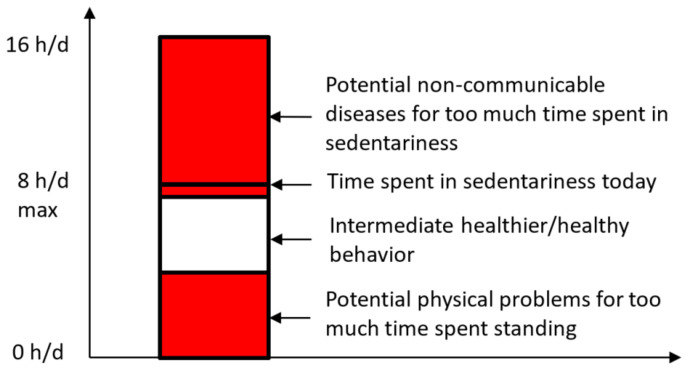
Time spent in sedentariness per waking day.

**Table 1 healthcare-09-00995-t001:** Objective time spent seated (using accelerometers and/or inclinometers).

Study	Objective Time Spent per Day in Sedentariness	Countries in Which the Study Was Performed
Smith et al. (2015)	10.6 h/d	England
Hamer, Coombs and Stamatakis (2014)	9.64 h/d	England
Loyen et al. (2017)	8.83 h/d	England, Norway, Portugal, and Sweden
Gibson, Muggeridge, Hughes, Kelly and Kirk (2017)	9.5 h/d	Scotland
Vallance et al. (2011)	8.5 h/d ^1^	Canada
Carson et al. (2014)	10.8 h/d	Canada
Healy, Matthews, Dunstan, Winkler and Owen (2011)	8.44 h/d	United States of America
Matthews et al. (2008)	8.53 h/d	United States of America

Note. h/d = hours per day; ^1^ The investigators measured 7.7 h/d of sedentariness in 13.9 h/d but they extrapolated their results to a waking day of 15.4 h).

## Data Availability

Not applicable.

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
