# Peer review of "Health Issues Due to the Global Prevalence of Sedentariness and Recommendations towards Achieving a Healthier Behaviour"

_healthcare, 2021, doi:10.3390/healthcare9080995_

Round 1

Reviewer 1 Report

The authors carefully and thoroughly responded to this reviewer's critique.  

Only a few minor suggestions:

Abstract, 3rd highlighted section (in yellow) - is "instead" the correct word?" Did you mean "indeed?"  Or "in addition?"

Introduction, 1st highlighted section, and Section 2, 1st highlighted section -  should be "It is important to mention that..."

Author Response

Comments and Suggestions for Authors

The authors carefully and thoroughly responded to this reviewer's critique.  

Only a few minor suggestions:

Abstract, 3rd highlighted section (in yellow) - is "instead" the correct word?" Did you mean "indeed?"  Or "in addition?"

Yes thank you. We replaced “Instead” by “In addition”.

Introduction, 1st highlighted section, and Section 2, 1st highlighted section - should be "It is important to mention that..."

We replaced “Important is to mention that...” by “It is important to mention that...”

Reviewer 2 Report

This is an opinion paper on the effects of sedentariness on human health.

Although the topic is interesting with important implications for human health, I feel that the presentation of the issue is done in a very abstract way, more like a compilation of several pieces of information presented in a simplistic way.

First of all, the text is in many parts very repetitive e.g. "Health problems associated with lack of physical activity have been thoroughly documented over the last few decades. Health problems associated with sedentariness have also been documented in the last decade." This part could easily be presented in a more concrete and scientifically sound way as :"Sedentariness and the lack of physical activity have been associated with health problems over the last decades". There are many parts like this throughout the text.

I would urge the authors to re-write several parts of the paper following a more scientific writing style and to avoid phrases like " In fact, the level of sedentariness is higher today than ever." 

Some statements need to be checked and referenced properly e.g. teleworking has advantages such as reduction of working hours, increase in the safety of employees. There are substantial counter-arguments against these statements, e.g. https://www.ilo.org/wcmsp5/groups/public/---dgreports/---ddg_p/documents/publication/wcms_791858.pdf (page 6 3rd paragraph).

Presenting some definitions used in the paper could be useful for example passive sedentariness, the context in which "modern day hunter gatherers" is used, sitting vs standing productivity, waking day.

I understand that some data is presented on the amounts of sedentariness from some published studies and some suggestions for action are made. Have the authors researched published information on interventions that have been applied in order to change the effects of sedentariness? It would be very interesting to see.  Also, apart from modern lifestyle promoting the sedentary lifestyle, are there other factors that lead to it, i.e age, family conditions, high position jobs, based on studies available?

Activity trackers can be used are available and can be used. Perhaps the authors can find some detailed relevant data on their effectiveness in changing the levels of sedentariness.

Moreover, specific suggestions should be made instead of urging the readers to discover them in reviews "We refer the reader to many other solutions presented in recent reviews to increase the time spent in the standing position".

Overall, the authors should reconsider the mode they use to convey a message to the readers and make it as clear as possible. Perhaps using a table to show the modes of time increase in the standing position could be better than the narrative way chosen. 

Finally, I do not find the analogy with the car on the tarmack road very helpful.

Author Response

Comments and Suggestions for Authors

This is an opinion paper on the effects of sedentariness on human health. Although the topic is interesting with important implications for human health, I feel that the presentation of the issue is done in a very abstract way, more like a compilation of several pieces of information presented in a simplistic way.

Thank you very much for the feedback. We are glad to read that the message is simple to read. At least, we succeeded in writing a complex synthesis in a comprehensive way. The reviewer is right that the presentation of the issue is done “in a very abstract way”. This was indeed our objective to do so this way. We could also have performed a review (with so many published studies) of each of the themes that we highlighted (see for example the devices used to measure sedentariness). Our choice was to make a general synthesis with the most recent published studies. The reader can then read any of the mentioned studies to go further in each direction.

In the general above feedback, you suggested “extensive editing of English language and style” which is difficult to understand as the manuscript was revised by a specific Editing Service (Editage). The reviewer also suggested that the introduction must be improved because it did not provide sufficient background and relevant references. Again, this suggestion is vague and rude regarding the usage of references as we used very recent reference (except perhaps new ones published in 2021). The reviewer suggested to improve 1) the research design, 2) the description of the methods and 3) the presentation of the results but these 3 aspects are not applicable, as suggested by reviewer # 2. However, following the reviewer’s comment, we improved the way to present some results in a Table. The reviewer suggests to improve the conclusion and we indeed removed the analogy that the reviewer disliked. Otherwise, the conclusion is just a summary of our opinion. We were careful about how to rewrite it.

First of all, the text is in many parts very repetitive e.g. "Health problems associated with lack of physical activity have been thoroughly documented over the last few decades. Health problems associated with sedentariness have also been documented in the last decade." This part could easily be presented in a more concrete and scientifically sound way as: "Sedentariness and the lack of physical activity have been associated with health problems over the last decades". There are many parts like this throughout the text.

We accept the remark that the text can be sometimes repetitive. We tried our best to reduce this issue in the updated manuscript. We agree that the two aforementioned sentences are long but they were actually not a repetition as in the first one it was written “over the last few decades” (with “s”) and the second time it is “in the last decade” (with no “s”). This distinction was important to suggest that “More light should be shed on sedentariness”. We turned the two sentences differently to shorten them and to avoid the problem raised by the reviewer. Furthermore, the introduction and discussion did not need so many details. For this reason:

  • We deleted the fact that “health problems associated with prolonged sedentariness are different, complementary, and additional to health problems associated with physical activity” as it is developed in the core of the manuscript.
  • Also in the introduction, we deleted 6 lines explaining various cases for various persons as it should be done in the core of the manuscript. In the introduction, we also deleted another sentence beginning by “In other words…” because this new sentence explained the same message differently
  • In the core of the manuscript, we repeated twice (involuntarily) the idea of alternating standing and sitting in two paragraphs. We combined this idea only in the appropriate paragraph.
  • In the introduction, we also removed the description of persons standing more or less (for example sitting for 10 h/d or sitting for 4 h/d). We described these examples twice in the core of the manuscript and now do it only in one single paragraph.
  • We completely rewrote the conclusion in removing the analogy and in writing a conventional conclusion.

We would still agree with the reviewer that even now, the message can still seem redundant. In fact, the first part (introduction) serves to alert what is coming, the second part (main core) serves to explain the main messages with details and reviews and the third part serve to be sure that the entire message is well understood. This is important to do so as even with some repetitions, the message may not be clear enough. As an example, the third reviewer  did not understand the entire message clearly, constraining us to repeat it again a third time in the conclusion. We were careful that the introduction and conclusion sections only repeat the most important message.

I would urge the authors to re-write several parts of the paper following a more scientific writing style and to avoid phrases like " In fact, the level of sedentariness is higher today than ever." 

We would be pleased to write the manuscript in a more scientific writing but the challenge is hard with only one suggestion. At least, the feedback from the reviewers does not concern the number of scientific references as we already cite 53 references in the present manuscript. Moreover, the references are extremely recent, with 48/53 references published in the last 10 years and 15/53 cited in the last 3 years (2019, 2020, 2021). An opinion is also a ‘scientific paper’, this is a synthesis of the literature reports.

Specifically for the sentence "In fact, the level of sedentariness is higher today than ever" we now write “In fact, the level of sedentariness is increasing worldwide at least since the middle of the 20th century." This is a fact with a convincing reference [Ng & Popkin, 2012]."

We searched to understand what is “less scientific” in our manuscript. At least we completely removed the analogy, it may help to become more scientific. Furthermore, we were particularly careful about removing useless repetitions (cf previous comment). Then, we thought that perhaps the reviewer did not appreciate some examples illustrated in the manuscript:  As mentioned in our earlier answer, these examples are not repeated three times but only discussed once to be clear that our general message then needs to be adapted to each person. We need to keep these explanatory examples for several reasons:

  • Before the second revision, the associate editor required to give some references to sustain these examples and we therefore added references which pointed a message going in the same direction. Hence, we cannot remove them as we extended them in the past revision.
  • Moreover, for this second revision, the reviewer # 2 suggested that our message was too much general and not enough specific to certain populations. For this reason, we had to explain in which way it can be understood by various people performing various types of motor activities during the day.

Please, be clearer on which part is written with “not enough scientific writing style” and we would be please to discuss/revise this part of the manuscript.

Some statements need to be checked and referenced properly e.g. teleworking has advantages such as reduction of working hours, increase in the safety of employees. There are substantial counter-arguments against these statements, e.g. https://www.ilo.org/wcmsp5/groups/public/---dgreports/---ddg_p/documents/publication/wcms_791858.pdf (page 6 3rd paragraph).

In checking the list of advantages suggested by Belzunegui-Eraso and Erro-Garcés (2020) and your suggested website, we agree that the “reduction of working hours” is not the best example of advantage of teleworking as it can be contested. Therefore, we removed this advantage in our list (although cited by Belzunegui-Eraso and Erro-Garcés, 2020 and actually true for many people). Thank you for noticing.

Presenting some definitions used in the paper could be useful for example “passive sedentariness”, the context in which "modern day hunter gatherers" is used, sitting vs standing productivity, waking day.

We agree that it is important to provide definitions and to be clear.

  • The notions of “passive sedentariness” or “passive sitting” were removed throughout in the manuscript as sedentariness is clearly discussed as passive (this is sitting). Now, “passive” is only used twice in the text to give an adjective for nonambulatory behaviors as these behaviors could be passive (such as when sitting) or more active (such as when standing without moving, squatting, kneeling).
  • The term “productivity” is now replaced by “work productivity” throughout the manuscript.
  • The term “waking day” is used here the time spent out of the bed during the day. We thought we did not need to define this term because its definition is obvious in its context. In the literature, this term is used in so many published manuscripts with no definition of what it means (e.g. in Chastin et al., 2018; Raichlen et al., 2020; Stamatakis et al., 2019; Vallance et al., 2011; WHO, 2020).
  • We do not understand which definition is requested for “the context in which modern day hunter gatherers is used”? Should we define what is a hunter-gatherer? In our manuscript, we referred to modern day hunter gatherers as “human living in a society in which most or all food is obtained by foraging — collecting or gathering wild plants and pursuing or hunting wild animals” (definition of Wikipedia). The term “modern-day” means that they are living today, this should be written in good English as this way of writing was suggested by Dr Fraser (Biotech Communication, Damery, France).

I understand that some data is presented on the amounts of sedentariness from some published studies and some suggestions for action are made. Have the authors researched published information on interventions that have been applied in order to change the effects of sedentariness? It would be very interesting to see.  Also, apart from modern lifestyle promoting the sedentary lifestyle, are there other factors that lead to it, i.e age, family conditions, high position jobs, based on studies available?

The reviewer is right that the objectives of the manuscript were to write a general synthesis discussing the prevalence of objective sedentariness, to discuss why the prevalence of sedentariness is so high, to highlight health problems associated with the high prevalence of sedentariness, and to discuss solutions to lower the prevalence of sedentariness. In the present manuscript, we highlighted that people need to stand more during the (waking) day. We explain pros and cons for spending more time in the standing position, we explained the balance to be found between sitting and standing. This is our important message and this is the novelty of the present manuscript. Then, the reviewer is right that many interventions exist to lower the prevalence of sedentariness. In our previous version of the manuscript, we already described some interventions that can be applied to reduce the time spent in sedentariness. We extended this paragraph in suggesting many interventions already described in the literature reports (see below this extended paragraph in the manuscript, two answers below the present one).

For the last question, our objective was to discuss and quantify a fact (sedentariness is very high) and to suggest solutions to lower this society issue (carefully standing more but not too much). We did not discuss specifically who is concerned with sedentariness. It is indeed a relevant question to know which factors could modulate the prevalence of sedentariness. This interesting question could be the objective of a full review, as this question would need many paragraphs and syntheses. We already discuss in our manuscript that people working in desk-based jobs are much more sedentary than other people. We can also highlight that watching TV at home is known to increase sedentariness (Guo et al., 2020; Ilakkuvan et al., 2019; Pinto Pereira et al. 2012). The prevalence of certain types of population (infants, students) have been studied (e.g. Castro et al., 2020) and the sedentary lifestyle has also been studied (e.g. Park et al., 2020; Prince et al., 2020). We think this is an interesting idea for a future manuscript, thank you for this suggestion.

Cited manuscripts:

  • Castro, O., Bennie, J., Vergeer, I., Bosselut, G., & Biddle, S. J. H. (2020). How Sedentary Are University Students? A Systematic Review and Meta-Analysis. Prevention Science: The Official Journal of the Society for Prevention Research, 21, 332‑343.
  • Guo, C., Zhou, Q., Zhang, D., Qin, P., Li, Q., Tian, G., … Hu, D. (2020). Association of total sedentary behaviour and television viewing with risk of overweight/obesity, type 2 diabetes and hypertension : A dose–response meta-analysis. Diabetes, Obesity and Metabolism, 22, 79‑90.
  • Ilakkuvan, V., Johnson, A., Villanti, A. C., Evans, W. D., & Turner, M. (2019). Patterns of Social Media Use and Their Relationship to Health Risks Among Young Adults. Journal of Adolescent Health, 64, 158‑164.
  • Park, J. H., Moon, J. H., Kim, H. J., Kong, M. H., & Oh, Y. H. (2020). Sedentary Lifestyle : Overview of Updated Evidence of Potential Health Risks. Korean Journal of Family Medicine, 41, 365‑373.
  • Pinto Pereira, S. M., Ki, M., & Power, C. (2012). Sedentary behaviour and biomarkers for cardiovascular disease and diabetes in mid-life : The role of television-viewing and sitting at work. PloS One, 7, e31132.
  • Prince, S. A., Roberts, K. C., Melvin, A., Butler, G. P., & Thompson, W. (2020). Gender and education differences in sedentary behaviour in Canada: An analysis of national cross-sectional surveys. BMC Public Health, 20, 1170.

Activity trackers are available and can be used. Perhaps the authors can find some detailed relevant data on their effectiveness in changing the levels of sedentariness.

The reviewer is right that many activity trackers exist. As we discussed, there are two types of trackers, 1) for science and 2) for the general population:

  • For science, many activity trackers exist and these devices accurately record the objective time spent in sedentariness. Researchers can obtain their data with software, they can analyze them to publish them. Many publications detail these trackers and their validity (e.g. Byrom, Stratton, Mc Carthy, & Muehlhausen, 2016; Feehan et al., 2018; Koster et al., 2016; Reid et al., 2017; Rosenberger, Buman, Haskell, Mcconnell, & Carstensen, 2016; Sanders et al., 2016; Steeves et al., 2015).
  • For the general population, it is not convenient to use these aforementioned trackers. However, many applications exist to reduce sedentariness (for example with mobile phone (Aldenaini, Alqahtani, Orji, & Sampalli, 2020; Buckingham, Williams, Morrissey, Price, & Harrison, 2019; Damen et al., 2020; Dunn, Gainforth, & Robertson-Wilson, 2018; Huang, Benford, & Blake, 2019; Low et al., 2020; Yamamoto et al., 2020), also smart swatches (Gill et al., 2018; Wang, König, & Reiterer, 2021). However, as we now added in the updated manuscript “both smart watches and mobile phones are limited as they do not measure and report the time spent seated during the day. Novel solutions are appearing in the literature but are still in development (Wang et al., 2021). Also, these devices do not measure sedentariness accurately because people do not carry their cell phone all day long. Moreover these devices are not perfectly accurate in their measure of the time spent in sedentariness. One solution was proposed by Gill et al. (2018) but this smart watch is still not commercially available today.”

Here are the afore-mentioned references (among so many other ones):

  • Byrom, B., Stratton, G., Mc Carthy, M., & Muehlhausen, W. (2016). Objective measurement of sedentary behaviour using accelerometers. International Journal of Obesity (2005), 40, 1809‑1812.
  • Feehan, L. M., Geldman, J., Sayre, E. C., Park, C., Ezzat, A. M., Yoo, J. Y., … Li, L. C. (2018). Accuracy of Fitbit Devices : Systematic Review and Narrative Syntheses of Quantitative Data. JMIR MHealth and UHealth, 6, e10527.
  • Koster, A., Shiroma, E. J., Caserotti, P., Matthews, C. E., Chen, K. Y., Glynn, N. W., & Harris, T. B. (2016). Comparison of Sedentary Estimates between activPAL and Hip- and Wrist-worn ActiGraph. Medicine and science in sports and exercise, 48, 1514‑1522.
  • Reid, R. E. R., Insogna, J. A., Carver, T. E., Comptour, A. M., Bewski, N. A., Sciortino, C., & Andersen, R. E. (2017). Validity and reliability of Fitbit activity monitors compared to ActiGraph GT3X+ with female adults in a free-living environment. Journal of Science and Medicine in Sport, 20, 578‑582.
  • Rosenberger, M. E., Buman, M. P., Haskell, W. L., Mcconnell, M. V., & Carstensen, L. L. (2016). Twenty-four Hours of Sleep, Sedentary Behavior, and Physical Activity with Nine Wearable Devices. Medicine & Science in Sports & Exercise, 48, 457‑465.
  • Sanders, J. P., Loveday, A., Pearson, N., Edwardson, C., Yates, T., Biddle, S. J. H., & Esliger, D. W. (2016). Devices for Self-Monitoring Sedentary Time or Physical Activity : A Scoping Review. Journal of Medical Internet Research, 18, e90.
  • Steeves, J. A., Bowles, H. R., McClain, J. J., Dodd, K. W., Brychta, R. J., Wang, J., & Chen, K. Y. (2015). Ability of Thigh-Worn ActiGraph and activPAL Monitors to Classify Posture and Motion. Medicine and science in sports and exercise, 47, 952‑959.

  • Aldenaini, N., Alqahtani, F., Orji, R., & Sampalli, S. (2020). Trends in Persuasive Technologies for Physical Activity and Sedentary Behavior : A Systematic Review. Frontiers in Artificial Intelligence, 0. https://doi.org/10.3389/frai.2020.00007
  • Buckingham, S. A., Williams, A. J., Morrissey, K., Price, L., & Harrison, J. (2019). Mobile health interventions to promote physical activity and reduce sedentary behaviour in the workplace : A systematic review. DIGITAL HEALTH, 5, 2055207619839883.
  • Damen, I., Brombacher, H., Lallemand, C., Brankaert, R., Brombacher, A., van Wesemael, P., & Vos, S. (2020). A Scoping Review of Digital Tools to Reduce Sedentary Behavior or Increase Physical Activity in Knowledge Workers. International Journal of Environmental Research and Public Health, 17, 499.
  • Dunn, E. E., Gainforth, H. L., & Robertson-Wilson, J. E. (2018). Behavior change techniques in mobile applications for sedentary behavior. DIGITAL HEALTH, 4, 2055207618785798.
  • Huang, Y., Benford, S., & Blake, H. (2019). Digital Interventions to Reduce Sedentary Behaviors of Office Workers : Scoping Review. Journal of Medical Internet Research, 21, e11079.
  • Low, C. A., Danko, M., Durica, K. C., Kunta, A. R., Mulukutla, R., Ren, Y., … Jakicic, J. M. (2020). A Real-Time Mobile Intervention to Reduce Sedentary Behavior Before and After Cancer Surgery : Usability and Feasibility Study. JMIR Perioperative Medicine, 3, e17292.
  • Yamamoto, K., Ebara, T., Matsuda, F., Matsukawa, T., Yamamoto, N., Ishii, K., … Kamijima, M. (2020). Can self-monitoring mobile health apps reduce sedentary behavior? A randomized controlled trial. Journal of Occupational Health, 62. https://doi.org/10.1002/1348-9585.12159

  • Gill, J. M. R., Hawari, N. S. A., Maxwell, D. J., Louden, D., Mourselas, N., Bunn, C., … Mutrie, N. (2018). Validation of a Novel Device to Measure and Provide Feedback on Sedentary Behavior. Medicine and Science in Sports and Exercise, 50, 525‑532.
  • Wang, Y., König, L. M., & Reiterer, H. (2021). A Smartphone App to Support Sedentary Behavior Change by Visualizing Personal Mobility Patterns and Action Planning (SedVis) : Development and Pilot Study. JMIR Formative Research, 5, e15369.

Moreover, specific suggestions should be made instead of urging the readers to discover them in reviews "We refer the reader to many other solutions presented in recent reviews to increase the time spent in the standing position".

Yes, the reviewer is right. Reviewer 2 also requested the same information. For this reason, we added a full paragraph discussing some of these interventions published elsewhere. The paragraph was extended as follows:

“Many other solutions exist to increase the time spent in the standing position.(Landais et al., 2020) Some examples of intervention techniques are “prompting”, “social influence”, “feedback”, and “anchoring”.(Landais et al., 2020; Michie, van Stralen, & West, 2011) Prompting refers to messages (posters, mobile phone messages, alert on a phone) engaging people to use the stairs instead of the elevators, to actively move to colleagues’ desk instead of emailing or calling them. Social influence refers to messages (oral, written) that specify what others have done (e.g. number of time other colleagues have taken the stairs during the day) to induce the same behaviors in all workers. Feedback refers to information of any behavior to find out if one respects expected results, either in front of personal objective or related to other’s ones. Anchoring refers to challenges suggested to workers, e.g. to assign a 5000 steps objective during the day or to stand more than 50 times during the day. Venema et al. (2018) discussed that most of the office workers were favorable about any intervention, even large ones, if these changes could improve their health. Only 11% disapproved the nudge intervention in their study.(Venema et al., 2018) In trains, buses, metros or when waiting at some places (for example in town for an appointment), people could stay in the standing position instead of sitting. In classrooms and/or universities, students could listen to professors sometimes upright, sometimes seated.(Finch, Tomiyama, & Ward, 2017) All these light changes would be appropriate to increase the time spent in the standing position to come back to a healthy/healthier intermediate behavior (Figure 1)”.

Overall, the authors should reconsider the mode they use to convey a message to the readers and make it as clear as possible. Perhaps using a table to show the modes of time increase in the standing position could be better than the narrative way chosen. 

We appreciate the suggestion of the reviewer. The idea of a table to show the prevalence of sedentariness is an excellent idea. For this reason, we added this table in the manuscript (cf. Table 1).

Finally, I do not find the analogy with the car on the tarmack road very helpful.

The analogy was removed and replaced by a traditional conclusion.

Reviewer 3 Report

Thank you for the invitation to review this manuscript. Whilst this paper addresses an important topic, in the form of sedentary behaviour, it provides no new insight into what is already known and has been published in other reviews. Whilst it is important to provide practical recommendations, those that the authors give are unsupported by research. Importantly they go against the recently published World Health Organisation guidance for sedentary behaviour. Furthermore, the frequent guidance that is provided instructing individuals to spend more time standing, ignores a wealth of research that has explored the impact of physical activity breaks from sitting on health markers. This guidance also is not inclusive for those who are unable to stand.

Introduction

  1. General: What do the authors mean by ‘health problems’? This is too vague and instead they should specify for example physical or mental health, or non-communicable diseases.
  2. Paragraph 1: The reference 4-6 do not seem the most appropriate to support the point the authors are making. For example, the Chastin et al (2018) study (reference 5) as not assessed any health parameter and instead validates a self-report tool. There are many, well cited reviews that summarise the evidence for the health risks associated with sedentary behaviour which the authors should instead cite
  3. Paragraph 1: This is not the correct definition of sedentary behaviour, the authors should instead cite the well published definition (Tremblay et al., 2017).
  4. Paragraph 1: ‘Import is to mention that’ does not make clear sense and this sentence requires restructuring

Section 2

  1. Paragraph 1: ‘By contrast, many humans…’ this sentence requires a reference to support this statement
  2. Paragraph 1: ‘8.5 h/d (after adjustment) in the United States’- after adjustment for what? This is not clear
  3. Paragraph 1: ‘these previous average values are underestimated because the participants did not wear the measurement device all day’. How do the authors know this? For data to be included from objective activity monitoring it must meet a certain wear time threshold. Therefore if the results are published in these studies, this must have been achieved. Participants may indeed have taken the monitors off to sleep, however this is not a sedentary behaviour
  4. Paragraph 1: ‘working in desk jobs spend even more time in sedentariness (from 65% to 82%, depending on the study).’ How does this value depend on the study when the authors then only reference one study, which is not a review/meta-analysis and therefore has not synthesised findings from multiple studies?
  5. Paragraph 2: ‘Moreover, it is typically assume’- this does not seem an appropriate phrase since if research has been conducted to show this it is not then an assumption
  6. Paragraph 2: ‘sitting behaviour is addictive’ the reference used here does not see to support this statement as it is a study assessing sitting on typing performance. The authors should review all references within the manuscript to ensure they are correct and appropriate.
  7. Paragraph 3: The opening to this paragraph is very long and reads like a continuous list. The authors should instead consider synthesising this information more succinctly and also referencing recent review/meta-analysis
  8. Paragraph 3: The authors discuss how sitting is independent from physical inactivity and that people need to move to interrupt their physical inactivity. But research has show moving to break up sitting is also effective. There have been many experimental studies exploring this, and these should be mentioned

Section 3

  1. General: Given the recently published WHO guidelines provide SB recommendations, these are not included or discussed.
  2. General: No discussion/information is provided about breaking up sitting, which a significant amount of SB literature has explored.
  3. Paragraph 1: The authors have created a 8h/d sitting threshold, however this is unsupported by any current evidence. Indeed, the recently published WHO guidelines for PA and SB describe how there is insufficient evidence to provide quantitative guidelines. How do the authors feel justified to go against the WHO guidelines?
  4. Paragraph 2: The authors describe that they are only aware of smart watches, however mobile applications are also available
  5. Paragraph 4: The authors recommendation to increase standing time is unsupported by literature. Furthermore, there was a recent editorial highlight the need for inclusivity in SB and PA messaging (Smith 2020). Standing may not be possible for all populations (e.g., due to those living with a disability). Therefore the authors are not providing inclusive, well-considered guidance.
  6. Paragraph 5: ‘In our study’- the authors are not reporting results from a study, please revise

Author Response

Comments and Suggestions for Authors

Thank you for the invitation to review this manuscript. Whilst this paper addresses an important topic, in the form of sedentary behaviour, it provides no new insight into what is already known and has been published in other reviews. Whilst it is important to provide practical recommendations, those that the authors give are unsupported by research. Importantly they go against the recently published World Health Organisation guidance for sedentary behaviour. Furthermore, the frequent guidance that is provided instructing individuals to spend more time standing, ignores a wealth of research that has explored the impact of physical activity breaks from sitting on health markers. This guidance also is not inclusive for those who are unable to stand.

Thank you very much for the comments/remarks that you provided. We are glad that you accepted our invitation to review the manuscript. As we know, the reviewer is very well aware of the published study in sedentariness, even the most recent ones. The reviewer is also very smart as we could judge from the comments and points of view. We find his initial comment extremely rude though. We agree that for someone well aware of the most recent literature, our manuscript may not seem novel (for example for a researcher in the WHO specifically studying this domain of research for years). However, many other people are not so much aware of the themes evoked, especially because they cross various domains of research. Furthermore, we need to highlight that, even for a specialist in sedentariness, our manuscript is novel in many respects:

  • The present manuscript is a very general synthesis. Usually in the literature reports, syntheses are not so general, they do not all together i) synthesize the prevalence of objective sedentariness, ii) discuss why the prevalence of sedentariness is so high, iii) write such a large list of health problems associated with the high prevalence of sedentariness (not exhaustive but still quite extended), and iv) discuss solutions to lower the prevalence of sedentariness. Just for the first point, we are not aware of any published manuscript reviewing specifically published reports focusing on the objective time spent in sedentariness. The reviewer is right that almost each of these sections could be found in various manuscript, but not together and not as much updated as in the present manuscript with several references published in 2020 and 2021.
  • Instead of saying only that “people should reduce their sedentariness”, as we can read in various manuscripts published in the literature, we state that “people should spend more time in the standing position” (either in nonambulatory or of course in ambulatory behaviors). This is not the same message. Indeed, on one hand, our message is positive, it gives some positive feedbacks about the standing position. We provide a solution (actually many solutions) to reduce sedentariness and the message is turned positively. In the literature, saying “people should reduce their sedentariness” is not a solution, it is turned in a negative way. Indeed suggesting to reduce the time spent in sedentariness does not mean that spending more time in the standing position is a good solution. Today, in the literature, it seems that there is no much solution to the problem of sedentariness, sedentariness seems to be a fatality. For sure we are very careful in explaining the limit of standing more during the day: we specifically alert that the time spent in the standing position should be increased very carefully as people need to avoid moving from prolonged sedentariness to prolonged standing. We are not aware of any synthesis putting so much light benefits - onto the standing position anywhere in the literature on sedentariness, on postural control, and/or on human factors engineering. Important is to consider that our claim is sustained by many studies discussing solutions/recommendations to stand a few minutes more a day (e.g. a call made in the standing position instead of when sitting). Please see our manuscript for more details.
  • To go even further in the aforementioned point, we explain pros and cons of spending more time in the standing position. This discussion is novel. This balance between sitting too much and standing too much has to be discussed in the literature. We are also not aware of any manuscript discussing the notion of “intermediate behavior”. This concept has not been used in the literature. To show even more the novelty of this concept and how to understand it, we created a Figure 1 showing what is meant by intermediate behavior. Thank you very much for suggesting that our message did not seem novel, as we now present our argument more strongly in the updated manuscript.
  • The discussion about task performance usually does not exist in manuscripts focusing on sedentariness. Also, the argument that task performance is as good when standing as when sitting can be known for researchers who fully read and know the literature (as you seemed to know) but this information is not highlighted in published reports focusing on non-communicable diseases related to sedentariness. We slightly increased the length of this paragraph to better explain its importance in our argumentation. It is an important novelty for the literature on sedentariness as it is a novelty also for the literature on postural control.
  • We are also not aware of any study discussing the stage of the art for wearable and accurate devices for the population, for practical use. For sure, many reviewers already discussed all kinds of devices (and many devices exist) but we are not aware of reviews specifically for user-friendly device that objectively measure their sedentariness very precisely. At least we provide a state of the art specifically for these devices.
  • Our manuscript discusses sedentariness vs. nonambulatory behaviors. This kind of discussion is not present in most of the manuscripts focusing on sedentariness. However, this is quite important as this is not the time spent in nonambulatory behaviors that seems to be problematic for non-communicable diseases but the time spent in (passive) sitting. We do not only explain that humans spend too much time in sedentariness but we explain that human physiology is not adapted to spend so much time in sedentariness. This explanation is based on researches performed with modern-day hunterer-gatherers. This explanation may be known by the reviewer but it is clearly not often written in the literature on sedentariness.

We can read the comment of the reviewer that our message “goes against the recently published WHO guidance for sedentary behaviour. Furthermore, the frequent guidance that is provided instructing individuals to spend more time standing, ignores a wealth of research that has explored the impact of physical activity breaks from sitting on health markers.” We clearly agree that spending 2 or 3 additional h/d in the standing position is surely not recommended (this is not our recommendation) but spending a few additional minutes/d in the standing position (this is our recommendation) would not hurt healthy adults. We read the WHO (2020) guidance carefully and it is not suggested anywhere that people should not spend some additional minutes per day in the standing position. As our recommendation (to stand more but only some additional min/d) was not clear in our manuscript, we added this suggestion even more, i.e. twice in the manuscript (in the introduction and in the conclusion sections).

For the last point, we agree with the reviewer that “This guidance also is not inclusive for those who are unable to stand”. Please see our answer to our comment # 17 for this comment.

Introduction

  1. General: What do the authors mean by ‘health problems’? This is too vague and instead they should specify for example physical or mental health, or non-communicable diseases.

Consistent with the reviewer, we updated the manuscript as follows:

  • Each time “health problem” was mentioned alone, we replaced “health problems” by non-communicable diseases”.
  • Each time, we wrote something like “Health problems associated with (prolonged) sedentariness” we did not modify the expression.

  1. Paragraph 1: The reference 4-6 do not seem the most appropriate to support the point the authors are making. For example, the Chastin et al (2018) study (reference 5) as not assessed any health parameter and instead validates a self-report tool. There are many, well cited reviews that summarise the evidence for the health risks associated with sedentary behaviour which the authors should instead cite

We agree with the reviewer that Chastin et al. (2018) was not the best reference. We replaced this reference by Levine (2015). Yes, we could cite many manuscripts here. However, we prefer citing only 3 references here because the previous sentence (focused on research on physical activity) also cited 3 references. We need to display the same number of references in both sentences to avoid the reader the bias that research on sedentariness may more important than research on physical activity.

  1. Paragraph 1: This is not the correct definition of sedentary behaviour, the authors should instead cite the well published definition (Tremblay et al., 2017).

The reviewer is right and we now do so.

  1. Paragraph 1: ‘Import is to mention that’ does not make clear sense and this sentence requires restructuring

Instead of “Important is to mention” we now write “It is important to mention that” as also suggested by reviewer 1.

Section 2

  1. Paragraph 1: ‘By contrast, many humans…’ this sentence requires a reference to support this statement

Many references are explained a few sentences later in the same paragraph. However, the reviewer is right that 1-2 references should be provided already here. So we cited 2 ones (Smith et al., 2015; Hadgraft et al., 2016).

  1. Paragraph 1: ‘8.5 h/d (after adjustment) in the United States’- after adjustment for what? This is not clear

The reviewer is right that our parenthesis did not help much. For this reason, we removed this parenthesis and inserted a note at the bottom of the page: “The investigators measured 7.7 h/d of sedentariness in 13.9 h/d but they extrapolated their results to a waking day of 15.4 hours)” This note is below Table 1 now.

  1. Paragraph 1: ‘these previous average values are underestimated because the participants did not wear the measurement device all day’. How do the authors know this? For data to be included from objective activity monitoring it must meet a certain wear time threshold. Therefore if the results are published in these studies, this must have been achieved. Participants may indeed have taken the monitors off to sleep, however this is not a sedentary behavior

Please, see the above answer. Investigators did not measure the time spent in sedentariness during the full waking day but “almost the full waking day”. For example, Matthews et al. (2008) only measured 13.9 h/d instead of the full waking day (15.4 h/d). The same limitation has been suggested by other investigators who measured the objective time spent in sedentariness. As the reviewer explained, we indeed never considered the time spent in bed.

  1. Paragraph 1: ‘working in desk jobs spend even more time in sedentariness (from 65% to 82%, depending on the study).’ How does this value depend on the study when the authors then only reference one study, which is not a review/meta-analysis and therefore has not synthesised findings from multiple studies?

The reviewer is right. To explain this issue, before submitting our original manuscript to HealthCare, we cited the three references (Gupta et al., 2016; Hadgraft et al., 2016; Parry & Straker, 2013). However, we thought that we could not cite more than 40 references in HealthCare and the only way to succeed in citing only 40 references was to delete here 2 of these 3 of the references. As we learned (on July 15th 2021) that we can cite as many references as we like in HealthCare, we now cite the three references. The reviewer will actually notice that we now have 53 references and not 40 anymore for this same reason.

  1. Paragraph 2: ‘Moreover, it is typically assumed’- this does not seem an appropriate phrase since if research has been conducted to show this it is not then an assumption

As researchers, we always need to be careful, especially in this case as this is not an absolute, incontestable, truth. In fact, the general assumption exists that “people perform better and are more productive when seated than when standing”. However, this general assumption is not exactly true. In fact, healthy adults are as efficient when standing than when sitting in various tasks (see three reviews of the literature: Drury et al., 2008; Karakolis & Callagha, 2014; Sui et al., 2019). So, this  afore mentioned sentence “Moreover, it is typically assumed that people perform better and are more productive when seated than when standing (Bergouignan et al., 2016)” is accurate. The sentence suggested by the reviewer “Moreover, people perform better and are more productive when seated than when standing (Bergouignan et al., 2016)” is also accurate in citing ref 13. However, this new sentence is not accurate at a more general level as people do not perform better when sitting than when standing (at least for very short tasks). Therefore, we could not and decided not to perform the requested change. This is simply a general intuition that we now better contradict in citing 2/3 of the afore mentioned references (Karakolis & Callagha, 2014; Sui et al., 2019).

  1. Paragraph 2: ‘sitting behaviour is addictive’ the reference used here does not see to support this statement as it is a study assessing sitting on typing performance. The authors should review all references within the manuscript to ensure they are correct and appropriate.

In their manuscript, Kar and Hedge (2016) wrote “Sitting in a chair is not bad in moderation, but in excess is addictive and harmful.” (in their introduction p. 460). However, we agree with the reviewer that Kar and Hedge (2016) is not the best reference to state this idea. Instead, Levine (2010) is the best reference. Originally, we knew this shortcoming but exactly for the same reason as just above, we had too many references (>> 40). We therefore had to remove Levine (2010). As HealthCare does not have any limit in references, this issue is now repaired as we cite Levine (2010).

  1. Paragraph 3: The opening to this paragraph is very long and reads like a continuous list. The authors should instead consider synthesising this information more succinctly and also referencing recent review/meta-analysis

We agree with the reviewer that the paragraph reads like a continuous list of disorders and diseases related to prolonged sedentariness. We deliberately wrote the paragraph this way in order to ‘afraid’ the readers about health problems related to sedentariness. We could indeed have synthetized this information more succinctly in only referring the readers to many published review/meta-analysis. However, in doing so, we would loosen the strength of the message that we want to share. We agree that this paragraph is long to read but the fact is to understand that problems related to sedentariness are huge and multiple. Levine (2015) already suggested a list of non-communicable diseases related to sedentariness as we did.

As for the remark that the review/meta-analysis could be more recent, the reviewer is extremely demanding. The references are extremely recent, with 48/53 references published in the last 10 years and 15/53 cited in the last 3 years (2019, 2020, 2021).

  1. Paragraph 3: The authors discuss how sitting is independent from physical inactivity and that people need to move to interrupt their physical inactivity. But research has shown moving to break up sitting is also effective. There have been many experimental studies exploring this, and these should be mentioned

The reviewer is absolutely right but paragraph 3 did not say that moving does not break sedentariness. In the sentence “Simply standing can interrupt sedentariness but cannot interrupt physical inactivity” we are not suggesting (at all) that only standing can disrupt sedentariness. To avoid this issue, we rephrased our sentence: “People need to stand and/or move to interrupt their sedentariness but simply standing cannot interrupt physical inactivity.” Later in this same paragraph 3, we even discuss that moving can reduce health problems associated with sedentariness more than simply standing. We agree with the reviewer that many experimental studies explore the way physical activity can help reducing physical activity but we already cite Ekelund et al. (2016) a famous research published in The Lancet. The problem raised by the reviewer was so important that we put a note after saying the first time that “people should spend more time in the standing position…” (please see in our updated introduction). We were careful about this issue throughout the manuscript because we did not only suggest that people should stand and stay in the quiet stance. Instead, we suggested that even just a standing position could reduce the time spent in sedentariness (among many other behavioral activities and/or physical activities). Thanks a lot for noticing.

Section 3

  1. General: Given the recently published WHO guidelines provide SB recommendations, these are not included or discussed.

The WHO (2020) should indeed be cited. It is now cited several times in our manuscript.

  1. General: No discussion/information is provided about breaking up sitting, which a significant amount of SB literature has explored.

We feel that the reviewer’s comment is too strong as we already discussed 3 ways to break up sitting. In fact, we already suggested to stand for 5 min every hour and/or to call someone or/or to hold stand-up meetings. However, the reviewer is right that the sentence “We refer the reader to many other solutions presented in recent reviews to increase the time spent in the standing position (Landais et al., 2020)” should be replaced by real recommendations. This remark was also mentioned by the other reviewer. For this reason, we replaced the previous sentence by the full coming paragraph:

“Many other solutions exist to increase the time spent in the standing position.(Landais et al., 2020) Some examples of intervention techniques are “prompting”, “social influence”, “feedback”, and “anchoring”.(Landais et al., 2020; Michie, van Stralen, & West, 2011) Prompting refers to messages (posters, mobile phone messages, alert on a phone) engaging people to use the stairs instead of the elevators, to actively move to colleagues’ desk instead of emailing or calling them. Social influence refers to messages (oral, written) that specify what others have done (e.g. number of time other colleagues have taken the stairs during the day) to induce the same behaviors in all workers. Feedback refers to information of any behavior to find out if one respects expected results, either in front of personal objective or related to other’s ones. Anchoring refers to challenges suggested to workers, e.g. to assign a 5000 steps objective during the day or to stand more than 50 times during the day. Venema et al. (2018) discussed that most of the office workers were favorable about any intervention, even large ones, if these changes could improve their health. Only 11% disapproved the nudge intervention in their study.(Venema et al., 2018) In trains, buses, metros or when waiting at some places (for example in town for an appointment), people could stay in the standing position instead of sitting. In classrooms and/or universities, students could listen to professors sometimes upright, sometimes seated.(Finch, Tomiyama, & Ward, 2017) All these small changes would be appropriate to increase the time spent in the standing position to come back to a healthy/healthier intermediate behavior (Figure 1).”

  1. Paragraph 1: The authors have created a 8h/d sitting threshold, however this is unsupported by any current evidence. Indeed, the recently published WHO guidelines for PA and SB describe how there is insufficient evidence to provide quantitative guidelines. How do the authors feel justified to go against the WHO guidelines?

We agree with the reviewer that every researcher and/or publication should be careful about this important aspect. However, the reviewer is also well aware that it is now time to move forward and to suggest a threshold (e.g. Chaput et al., 2020: in the title = “a provisional benchmark is better than no benchmark at all”). There are many advantages to suggest a threshold (e.g. Chaput et al., 2020). Yes, we are aware that it is an extremely difficult task and that the WHO or other publications still did not do suggest such a threshold (unfortunately). We say “unfortunately” for many reasons:

  • Without any threshold, people will never feel concerned by prolonged sedentariness as we discuss and as Chaput et al. (2020) also discussed. Without any threshold, nobody will change his/her behavior and nobody will fill concerned about the message. Personally, when I (Dr Bonnet) read the WHO (2020) I do not feel concerned about the recommendations for sedentary behavior while in reading our present manuscript, I feel extremely concerned (as I spend > 10 h/d working on my computer).
  • Without any threshold, it is impossible to give feedback to the population whether they overpassed this threshold (as it does not exist). This is a real problem.
  • Without any threshold, it is impossible to decide – as a researcher – whether some people (or even the general populations in some countries) have overpassed this limitation (as the limitation does not exist).

For sure, we are aware that it is not possible, still today, to claim a perfectly accurate threshold of sedentariness. In the present manuscript, we do not suggest a threshold but we mention a general baseline (8 h/d) already used in several published manuscripts. We need to mention that these investigators chose 6 to 8 h/d as thresholds in their analysis, we did not choose 6 to 8 hours from nowhere. For example, Ekelund et al. (2016) in The Lancet discussed many breaking durations per day of time spent seated (< 4 h/d, < 6 h/d, < 8 h/d). We simply report their choice and analyze which duration could become a first conservative baseline threshold to move forward in science. As we suggest in our manuscript this is not really a question of suggesting or finding the perfect threshold that matters but to build on existing findings to adjust this threshold with future studies. The same procedure was used with normative values for physical activity that have been validated in the new WHO (2020). Without any threshold, science is stuck and does not move forward.

The reviewer suggested that this 8 h/d threshold is unsupported by any current evidence. However, we cited 6 references showing at least that people can be affected by non-communicable disease if they spend more than 8 h/d in sedentariness. Our review of the literature showed that all people who measured the objective prevalence of sedentariness found > 8 h/d in the last 12 years. For this reason, our baseline value (8 h/d) is an initial conservative baseline threshold based on the literature reports. Our threhold is also supported by the fact that modern-day hunter-gatherers are almost not affected by non-communicable diseases (Raichlen et al., 2020). We are aware that many other factors are different between hunter-gatherers and people living in modern societies, but this is an additional indirect validation going int the same direction. Also, a review of the literature showed in 2010 that many aspects – including sedentariness – were increasing the burden of non-communicable disease (Habib & Saha, 2010). We agree that the “burden of non-communicable disease” discussed by Habib and Saha (2010) does not only comes from sedentariness, but these authors suggested that sedentariness is one of the cardinal factors for non-communicable diseases. This information is also an additional point consistent with our synthesis that 8 h/d could be a conservative initial threshold for the harmful effect of sedentariness. So much evidence is going in the same direction that we need to move forward. We need to do so carefully and we hope that this is what we did.

  1. Paragraph 2: The authors describe that they are only aware of smart watches, however mobile applications are also available

Yes, the reviewer is right. Yes, this is important to signal that mobile applications also exist, thanks a lot for noticing. We provide references for such mobile applications. The manuscript was updated and can be read as follows: “Today, we are aware of smart watches and mobile applications to signal if individuals spend too much time on sedentariness (Aldenaini et al., 2020; Beckwith, 2021). This is a good start to break prolonged sedentariness, to suggest standing and moving around, or to begin a short period of time working or performing other tasks while in the standing position. However, both smart watches and mobile phones are limited as they do not measure and report the total time spent seated during the day. Novel solutions are appearing in the literature but are still in development (Wang et al., 2021) These devices do not measure sedentariness accurately because people do not carry their cell phone all day long. Moreover these devices are not perfectly accurate in their measure of the time spent in sedentariness. One solution was proposed by Gill et al (2018) but this smart watch is still not commercially available today”

  1. Paragraph 4: The authors recommendation to increase standing time is unsupported by literature. Furthermore, there was a recent editorial highlight the need for inclusivity in SB and PA messaging (Smith 2020). Standing may not be possible for all populations (e.g., due to those living with a disability). Therefore the authors are not providing inclusive, well-considered guidance.

We know that the reviewer is well aware of the literature reports in sedentariness. However, with respect to the first sentence, many investigators already discussed and demonstrated the benefits of standing a few minutes more here or there during the day (see our manuscript to find these references). We do not recommend to spend 2 or 3 additional h/d in the standing position, surely not. We are recommending to spend some additional minutes per day in the standing position. Yes, our message does not concern everybody. Our message is relative to everyone lifestyle habits and conditions. We already explained this point of view in various sentences in our manuscript. For example, we wrote that “Considering a waking day of 16 h/d, any one spending more than 10 h/d in the standing position does not need to increase the time spent in the standing position.” AND “a person who stands for only 4 h/d would clearly need to increase the time spent in the standing position and/or dynamic nonambulatory behaviours.” AND “in order for person B used to stand 7 h/d every day, no change in lifestyle habits would be required, as this individual would already use the intermediate behavior.” We did not mention that people not able to stand (those living with a disability) were not concerned with this message but we added this information in the updated manuscript, in the introduction, in the middle of the manuscript and in conclusion as follows:

  • Introduction: “We need to be clear that our message is general as we only discuss the total amount of time spent seated per day without differentiating the context in which people sit (e.g. at work, at home in front of the TV…). Moreover, the recommendations are general as they are not specific to any group of individuals (children, adults, older adults, individuals with impairment)”
  • Middle of the manuscript: “It should be adapted to people with disabilities and/or difficulties to stand.”
  • In conclusion: “One limitation is that our message is general, too general, as we did not precise any specific minimum threshold for each type of people (infants, healthy adults, older adults, persons with disabilities) and/or for individuals in particular. Future researches should carefully analyse, display and then adjust these respective minimum thresholds because the population needs to be informed and aware of these thresholds to be active in preventing, at least limiting, non-communicable diseases related to prolonged sedentariness.”

With respect to “Smith (2020)”, the reviewer was unclear which reference was cited. In searching with key words “physical activity”, “sedentary behavior”, “editorial highlights”, “smith 2020” we found “Meyer et al. (2020): Changes in Physical Activity and Sedentary Behavior in Response to COVID-19 and Their Associations with Mental Health in 3052 US Adults” with Dr Smith as a co-author. In this manuscript, there is no mention about the recommendation to changing the time spent in the standing behavior. If the reviewer could be clearer about the reference to read, we will be careful about recommendations that do not sustain our ones.

  1. Paragraph 5: ‘In our study’- the authors are not reporting results from a study, please revise

We replaced “In our study” by “In our manuscript”.

Round 2

Reviewer 2 Report

This is a much better version of the manuscript, which is now more structured and comprehensive. Thanks to the authors who have put a lot of work in improving the document. I believe it can now be published as is, but if the authors choose to follow on with this topic in the future, focusing on specific groups and making specific suggestions will be better.

Author Response

Thank you very much for this positive feedback and advice for future research. We agree that future researches should focus on specific groups, this is a very interesting topic, thanks.

Reviewer 3 Report

Thank you for the invitation to review this revised manuscript. The authors have taken the time to respond to the reviewer comments and amend the paper accordingly. It is pleasing to see the recommendations surrounding guidelines for sitting and standing have been toned down in places. However, further changes are needed to improve the manuscript as described in detail below:

It is also disappointing that the authors describe myself as ‘rude’, when myself and the others reviewers have taken a lot of time to provide detailed, constructive feedback to improve their submission and allow their work to be published. Secondly, it is disappointing the authors refer to myself as ‘he’, assuming it will be a male who is reviewing a scientific paper.

Introduction

  1. Lines 27-30: This added sentence does not fit with the flow of this paragraph. The authors have just introduced that a lack of PA is associated with diseases, and have not been discussing standing. Also, this sentence is not easy to follow. The authors should instead just use a phrase such as ‘spending more time standing or performing physical activity’. It is not easy or obvious for the reader to remember that the term standing actually refers to ‘spending more time in the standing position and doing any physical activities (simply standing, walking, performing any kind of physical activity…’
  2. Line 31: The authors are still using the phrase ‘health problems’, please revise
  3. Lines 32-34: A definition of ‘sitting’ as outlined by Tremblay et al., 2017 has now been added. However, throughout the manuscript, and indeed in this sentence, the term ‘sedentariness’ is used. The definition of Sedentary behavior is: ‘any waking behavior characterised by an energy expenditure ≤1.5 metabolic equivalents (METs), while in a sitting, reclining or lying posture’. The authors should review the terminology used throughout the manuscript and ensure the terms/definitions are used for the behaviours there are referring to.
  4. Lines 46-48: By stating that doing any PA in the standing stance, the authors are not considering research that has explored the use of pedal workstations as a means to reduce sitting time. This is conducted still while seated, yet is a form of PA
  5. Line 50: ‘a well-documented dangerous behaviour’- this statement should be supported by a reference

Section 2

  1. Figure 1, Line 88: ‘inn’ is spelt incorrectly, please revise
  2. Table 1: The Vallance study include the superscript 2, however this is not linked to the footnote.
  3. Line 77-78: In the authors responses they explained the reason for the statement ‘did not wear the measurement device all day’. However this still is not clear in the manuscript what this means. Instead, the authors should consider providing more specific details, such as ‘participants did not wear the device during all waking hours’. This would remove any ambiguity with the term ‘all day’ which might suggest you are considering the whole 24hr day, which would therefore include sleep
  4. Paragraph 3: My previous comment was that the authors should reference recent review/meta-analyses on this topic, it was not that the current references were outdated, however by included meta-analyses it would synthesis what is known. Also, by design, referencing a meta-analysis shows that there is a range of existing research for the topic for a meta-analysis to be conducted. This should further create the ‘afraid’ factor the authors describe in their response. I would recommend the authors look at recent meta-analyses that have assessed endothelial function, blood glucose, insulin and lipid levels (Patterson et al., 2020, doi: 10.1007/s40279-020-01325-5; Loh et al, 2020, doi: 10.1007/s40279-019-01183-w.).
  5. Line 136: the phrase ‘little by little disrupted’ is too informal for a scientific publication, please revise
  6. Lines 137-139: These sentences are confusing and appear contradictory. First it is said people need to stand and/or move to interrupt SB, then it is said standing is not enough, then that people need to move. Please revise.

Section 3

  1. Line 183: What does ‘cf’ represent? Please revise
  2. Line 219-220: I see that the authors are including non-ambulatory or ambulatory standing behaviour to try encompass standing and activity, however this is not that clear to read. Also, as previously mentioned, it does not consider interventions that allow participants to be active but whilst still seated, such as pedal workstations
  3. Lines 223-224: The authors should reference this paper: Smith et al. (2002) Disability, the communication of physical activity and sedentary behaviour, and ableism: a call for inclusive messages, bjsports-2020-103780.
  4. Lines 224-228: These messages need to be toned down. Previously the authors have been more conservation with their guidelines, but now they are stating that anyone spending less than 6hrs a day sedentary, does not need to increase their standing time. There is no evidence to support this, so it cannot be stated so definitely. That same apples for the statement that behavioural habits should only change for people who spend less than 8hrs day standing
  5. Lines 238-241: The authors define an intermediate behaviour by using the term intermediate behaviour in the explanation, which is not clear, please revise
  6. Lines 256-257: The highlighted sentence does not make clear sense, please revise
  7. Lines 263-264: ‘The important message is not that alternating between standing and sitting is good for health as such, but that it is a mean to an end.’. This statement goes against research showing the benefit of frequent breaking up sitting on marker of health.

Conclusion

  1. Line 314-315: ‘he development of sit-stand desks’- these are already developed and commercially available, please revise.
  2. Line 321- The start of this sentence does not make sense, please revise
  3. Line 323: Please amend the phrase ‘Future researches’

Author Response

Comments and Suggestions for Authors

Thank you for the invitation to review this revised manuscript. The authors have taken the time to respond to the reviewer comments and amend the paper accordingly. It is pleasing to see the recommendations surrounding guidelines for sitting and standing have been toned down in places. However, further changes are needed to improve the manuscript as described in detail below:

Thank you for this positive introductive comment that we appreciate.

It is also disappointing that the authors describe myself as ‘rude’, when myself and the others reviewers have taken a lot of time to provide detailed, constructive feedback to improve their submission and allow their work to be published. Secondly, it is disappointing the authors refer to myself as ‘he’, assuming it will be a male who is reviewing a scientific paper.

For the first point, the reviewer is right. Important is to thank all three of you (you and the two reviewers) to have taken so much time and effort to help us improving the manuscript. You provided so many relevant feedback/instructions/advices that we need to thank you so much. We used the term ‘rude’ only with respect to your comment suggesting that nothing was new in our manuscript. We provided a list of bullets showing that this comment was exaggerated. For the second point, we need to apologize as we made a mistake. To explain that mistake, we invited a specific person (a male) to be a reviewer and so many aspects of your comments matched with this person’s background and knowledge that we thought you could be that person. The important point is that your comments were and are again very smart, you are a brilliant researcher, whoever you are. So, very sorry again for the confusion.

Introduction

  1. Lines 27-30: This added sentence does not fit with the flow of this paragraph. The authors have just introduced that a lack of PA is associated with diseases, and have not been discussing standing. Also, this sentence is not easy to follow. The authors should instead just use a phrase such as ‘spending more time standing or performing physical activity’. It is not easy or obvious for the reader to remember that the term standing actually refers to ‘spending more time in the standing position and doing any physical activities (simply standing, walking, performing any kind of physical activity…’

We are sorry for this issue caused by the fact that HealthCare does not allow footnotes (these parentheses was planned to be a footnote located at the bottom of the page). Sorry, we did not see that our footnote came into the text (and not as a footnote) when submitting our updated manuscript.

In the 2nd paragraph of the manuscript we explain the definition provided in these parentheses (lines 27-30). Hence, we did not need to keep these aforementioned parentheses (lines 27-30) anymore. Therefore, we removed these lines.

  1. Line 31: The authors are still using the phrase ‘health problems’, please revise

In our previous answer, we explained that:

  • Each time “health problem” was mentioned alone, we replaced “health problems” by non-communicable diseases”.
  • Each time, we wrote something like “Health problems associated with (prolonged) sedentariness” we did not modify the expression.

As the reviewer is still concerned, we replaced the term health problems by non-communicable diseases even when we wrote “health problems was associated with (prolonged) sedentariness”. We did it 6 times throughout the manuscript.

  1. Lines 32-34: A definition of ‘sitting’ as outlined by Tremblay et al., 2017 has now been added. However, throughout the manuscript, and indeed in this sentence, the term ‘sedentariness’ is used. The definition of Sedentary behavior is: ‘any waking behavior characterised by an energy expenditure ≤1.5 metabolic equivalents (METs), while in a sitting, reclining or lying posture’. The authors should review the terminology used throughout the manuscript and ensure the terms/definitions are used for the behaviours there are referring to.

Yes, the reviewer is right that we should better use this definition. We now define sedentariness exactly as the reviewer suggested except that we did not write ‘METs’ as we never use this term in the manuscript.

  1. Lines 46-48: By stating that doing any PA in the standing stance, the authors are not considering research that has explored the use of pedal workstations as a means to reduce sitting time. This is conducted still while seated, yet is a form of PA

The reviewer is completely right. We were thinking about this aspect but did not want to include it in the manuscript to avoid confusion. Indeed, in the previous version of the manuscript, we had not mentioned that energy expenditure should be lower than 1.5 METs, which was indeed something lacking in our manuscript (as suggested in the previous comment # 3). As performing physical activity while sitting (indeed) cannot be considered as sedentariness, we updated our definition. We now write “…spending more time in the standing position (simply standing, walking, performing any kind of other motor action) or doing any physical activity even when sitting (e.g. cycling, rowing in a boat) during the day is a good solution to spending less time in sedentariness”.

  1. Line 50: ‘a well-documented dangerous behaviour’- this statement should be supported by a reference

Yes, the reviewer is right: we provide two references (these references are discussed later in the manuscript but indeed should already be cited here)

Section 2

  1. Figure 1, Line 88: ‘inn’ is spelt incorrectly, please revise

Thank you for noticing: “inn” was replaced by “in”

  1. Table 1: The Vallance study include the superscript 2, however this is not linked to the footnote.

The problem is the same as for the 1st footnote: HealthCare does not allow footnotes. The text for this footnote existed but was put in the Note below the table indeed with no superscript 2. As footnotes are not allowed, we put a superscript and indeed write our message in the footnote with a corresponding superscript.

  1. Line 77-78: In the authors responses they explained the reason for the statement ‘did not wear the measurement device all day’. However this still is not clear in the manuscript what this means. Instead, the authors should consider providing more specific details, such as ‘participants did not wear the device during all waking hours’. This would remove any ambiguity with the term ‘all day’ which might suggest you are considering the whole 24hr day, which would therefore include sleep

Yes indeed, thanks for noticing: we replaced ‘all day’ by ‘waking hours’.

  1. Paragraph 3: My previous comment was that the authors should reference recent review/meta-analyses on this topic, it was not that the current references were outdated, however by included meta-analyses it would synthesis what is known. Also, by design, referencing a meta-analysis shows that there is a range of existing research for the topic for a meta-analysis to be conducted. This should further create the ‘afraid’ factor the authors describe in their response. I would recommend the authors look at recent meta-analyses that have assessed endothelial function, blood glucose, insulin and lipid levels (Patterson et al., 2020, doi: 10.1007/s40279-020-01325-5; Loh et al, 2020, doi: 10.1007/s40279-019-01183-w.).

Thank you very much for the two important publications. We included both Paterson et al. (2020) and Loh et al. (2020) in our manuscript, as suggested.

  1. Line 136: the phrase ‘little by little disrupted’ is too informal for a scientific publication, please revise

Yes indeed, we replaced ‘little by little’ by ‘over time’

  1. Lines 137-139: These sentences are confusing and appear contradictory. First it is said people need to stand and/or move to interrupt SB, then it is said standing is not enough, then that people need to move. Please revise.

The sentence “People need to stand and/or move to interrupt their sedentariness but simply standing cannot interrupt physical inactivity” was not contradictory. Indeed, we discussed sedentariness first and physical inactivity second. As stated in the previous sentence “Sedentariness differs from physical inactivity”, so we could expect to read a difference between sedentariness and physical inactivity. To improve the manuscript, we removed this sentence because we could add the precision in the following one. We now write: “Indeed, people need to move to interrupt their physical inactivity while simply standing can already interrupt sedentariness.”

Section 3

  1. Line 183: What does ‘cf’ represent? Please revise

Indeed, ‘cf’ was removed

  1. Line 219-220: I see that the authors are including non-ambulatory or ambulatory standing behaviour to try encompass standing and activity, however this is not that clear to read. Also, as previously mentioned, it does not consider interventions that allow participants to be active but whilst still seated, such as pedal workstations

We agree that spending more time in physical activity – even when sitting – is a good way to avoid non-communicable diseases caused by sedentariness. However, many other investigators already discussed this aspect and we now do so earlier in the manuscript. Furthermore, the paragraph pointed out by the reviewer specifically explains pros and cons of the standing position, it highlights the need for people to stand more in being careful to avoid health problems associated with prolonged standing. For both reasons, we cannot and should not include what the reviewer refers to in this paragraph of the manuscript.

Sorry but we do not understand what is difficult to understand in the sentence: “In general, people do not spend enough time in the standing position either in nonambulatory or ambulatory behaviours”? We do not know how to improve clarity of the sentence especially because the following sentence read as follows: “Therefore, our recommendation is to find solutions to spend more time in the standing position during the day.” For these reasons, we did not change anything here.

  1. Lines 223-224: The authors should reference this paper: Smith et al. (2021) Disability, the communication of physical activity and sedentary behaviour, and ableism: a call for inclusive messages, bjsports-2020-103780.

Very nice manuscript that we indeed cited as requested. This is amazing and we appreciate very much your knowledge and awareness of so recent studies and, more importantly, sharing such knowledge with us!

  1. Lines 224-228: These messages need to be toned down. Previously the authors have been more conservation with their guidelines, but now they are stating that anyone spending less than 6hrs a day sedentary, does not need to increase their standing time. There is no evidence to support this, so it cannot be stated so definitely. That same apples for the statement that behavioural habits should only change for people who spend less than 8hrs day standing

We completely agree with the reviewer, thank you very much for noticing. We replaced “any one spending less than 6 h/d in sedentariness does not need to increase the time spent in the standing position” by “any one spending less than 6 h/d in sedentariness may not need so much to increase the time spent in the standing position”. And we replaced “Behavioural habits should only change in people who spend less than 8 h/day in the standing position” by “Behavioural habits should primarily change in people who spend less than 8 h/day in the standing position”

  1. Lines 238-241: The authors define an intermediate behaviour by using the term intermediate behaviour in the explanation, which is not clear, please revise

Yes, we agree with the reviewer. We replaced “People who have an intermediate behaviour could be referred to as people who do not engage in prolonged sedentariness and/or prolonged standing but who adopt an intermediate behaviour” by “People who have an intermediate behaviour could be referred to as people who are not engaged in prolonged sedentariness and/or prolonged standing but who still spend enough time in both behaviours during the waking day”

  1. Lines 256-257: The highlighted sentence does not make clear sense, please revise

The sentence “For health concern, it is better to get away of the threshold” was removed.

  1. Lines 263-264: ‘The important message is not that alternating between standing and sitting is good for health as such, but that it is a mean to an end.’. This statement goes against research showing the benefit of frequent breaking up sitting on marker of health.

Yes, the sentence was inappropriate. For this reason, this sentence “The important message is not that alternating between standing and sitting is good for health as such, but that it is a mean to an end” was removed. Thank you very much for pointing this out.

Conclusion

  1. Line 314-315: ‘the development of sit-stand desks’- these are already developed and commercially available, please revise.

Yes, sit-stand desks already exist since a long time and are well available commercially. We replaced “the development of sit-stand desks” by “more use of sit-stand desks”.

  1. Line 321- The start of this sentence does not make sense, please revise

We replaced “One limitation of our manuscript is that our message was general as we did not provide any specific minimum threshold for each type of people (infants, healthy adults, older adults, persons with disabilities) and/or for individuals in particular” by “In our manuscript, we are aware that we did not provide any specific minimum threshold for each type of people (infants, healthy adults, older adults, persons with disabilities) and/or for any individual in particular.”

  1. Line 323: Please amend the phrase ‘Future researches’

We do not understand well the problem here. We replaced ‘Future researchers’ by ‘Future studies’.